# Development of Novel Monoclonal Antibodies to Wheat Alpha-Amylases Associated with Grain Quality Problems That Are Increasing with Climate Change

**DOI:** 10.3390/plants12223798

**Published:** 2023-11-08

**Authors:** Amber L. Hauvermale, Courtney Matzke, Gamila Bohaliga, Mike O. Pumphrey, Camille M. Steber, Andrew G. McCubbin

**Affiliations:** 1Department of Crop and Soil Sciences, Washington State University, Pullman, WA 99164, USA; ahauvermale@wsu.edu (A.L.H.); bohaligag@gmail.com (G.B.); m.pumphrey@wsu.edu (M.O.P.); 2School of Biological Sciences, Washington State University, Pullman, WA 99164, USA; 3Wheat Health, Quality and Genetics Unit, United States Department of Agriculture-Agricultural Research Service, Pullman, WA 99164, USA

**Keywords:** wheat, alpha-amylase, monoclonal antibodies, immunoassay, enzyme-linked immunosorbent assay, preharvest sprouting, end-use quality, falling number

## Abstract

Accurate, rapid testing platforms are essential for early detection and mitigation of late maturity α-amylase (LMA) and preharvest sprouting (PHS) in wheat. These conditions are characterized by elevated α-amylase levels and negatively impact flour quality, resulting in substantial economic losses. The Hagberg–Perten Falling Number (FN) method is the industry standard for measuring α-amylase activity in wheatmeal. However, FN does not directly detect α-amylase and has major limitations. Developing α-amylase immunoassays would potentially enable early, accurate detection regardless of testing environment. With this goal, we assessed an expression of α-amylase isoforms during seed development. Transcripts of three of the four isoforms were detected in developing and mature grain. These were cloned and used to develop *E. coli* expression lines expressing single isoforms. After assessing amino acid conservation between isoforms, we identified peptide sequences specific to a single isoform (TaAMY1) or that were conserved in all isoforms, to develop monoclonal antibodies with targeted specificities. Three monoclonal antibodies were developed, anti-TaAMY1-A, anti-TaAMY1-B, and anti-TaAMY1-C. All three detected endogenous α-amylase(s). Anti-TaAMY1-A was specific for TaAMY1, whereas anti-TaAMY1-C detected TaAMY1, 2, and 4. Thus, confirming that they possessed the intended specificities. All three antibodies were shown to be compatible for use with immuno-pulldown and immuno-assay applications.

## 1. Introduction

Wheat α-amylase enzymes play a critical role in starch mobilization during seed development and seed germination [1]. However, the accumulation of abnormally high-levels of α-amylase enzymes during grain development, or in mature grain, negatively impacts wheat end-use quality [2,3]. Elevated α-amylase is a world-wide problem in wheat production caused by either of two distinct genetic phenomena, preharvest sprouting (PHS) and late maturity α-amylase (LMA) (reviewed by [4,5,6]). It is important to detect elevated α-amylase activity in mature grain because it results in increased risk of poor end-product quality, such as cakes that fall and sticky bread or noodles (reviewed in [2,7]). For almost 60-years the wheat industry has relied on the Hagberg–Perten Falling Number (FN) assay to assess α-amylase activity in wheatmeal [8,9,10]. The FN method is based on the effect of α-amylase on the viscosity of a heated wheatmeal/water mixture or gravy. The result of the FN method is a falling number. Which is the number of seconds that it takes a stir bar to fall a defined distance through the wheatmeal gravy. An FN of <300 s is regarded as unacceptable in downstream product applications (bread, cakes, noodles) and leads to significant discounting of crop value.

Alpha-amylase can be expressed both during grain development and germination [4,6,11,12]. Preharvest sprouting is the initiation of mature grain germination on the mother plant when rain occurs before harvest [13,14]. During germination, the plant hormone gibberellin A (GA), produced by the germinating embryo, induces α-amylase expression in the aleurone cell-layer surrounding the endosperm [15,16]. Alpha-amylase digestion of starch mobilizes carbohydrates for use by the germinating seedling. GA treatment can induce α-amylase in isolated aleurones, enabling production of fairly pure germination-induced α-amylases [17]. Alpha-amylase is expressed during early grain development and then decreases during grain maturation [12,18,19]. LMA susceptible wheat varieties express α-amylases during the late maturation phase of seed development in response to cooler temperatures [4,20]. However, PHS results in higher levels of α-amylase that may have a more profound effect on end-product quality than LMA [3,21,22,23].

Though the FN method has been the industry standard amylase assay for many years, it is far from ideal, as it is fraught with technical issues and is costly and time consuming [7,24]. Commercially available alternatives to FN exist. The Rapid Visco Analyzer (RVA) is a more accurate and informative alternative to FN that also measures the effect of α-amylase on flour pasting properties [24]. However, the RVA instrument is expensive, and the approach has a lower throughput [2,7]. Since FN measures α-amylase activity based on decreasing viscosity, it is influenced by other factors affecting viscosity including protein levels and by starch properties (reviewed by [10]). Direct measurement of α-amylase requires either immunological detection of the protein or measurement of enzyme activity [7]. Enzyme assays are routinely used for evaluating the level of α-amylase enzyme activity in milled flour and have been modified to accommodate high-throughput screening platforms in academic settings [3,25,26,27]. However, the disadvantage of these tests is that small variations in sample pH, incubation time, or incubation temperature affect enzyme activity and can result in run-to-run variation. These sources of variation make α-amylase enzyme testing platforms challenging to standardize across testing years and locations. Furthermore, like the FN test, enzyme assays are unable to distinguish between independent isozymes known to be differentially expressed during LMA versus PHS [19].

Previous work showed that an α-amylase immunoassay might be a good alternative to FN. The WheatRite immunoassay used antibodies to wheat α-amylases and was more accurate, faster to run, cheaper, and easier to use in diverse testing locations (in the field, at grain elevators, or in laboratories) than the aforementioned alternatives [28,29]. Despite these attributes, WheatRite was ultimately pulled from the global market [28,29]. A less than ideal aspect of the WheatRite assay was that it used a combination of polyclonal and monoclonal antibodies [29]. The polyclonal antibodies employed were raised to a spectrum of (high and low isoelectric point [pI]) amylase isoforms and the monoclonal antibody specific to high-pI isoforms [29]. The use of polyclonal antibodies in a commercial diagnostic kit is problematic, as they must be continuously generated by immunizing new animals. Each of which may react differently to the antigen, leading to between batch variation. Currently, a 96-well ELISA using these antibodies is available, but only for research and only in Australia [20]. The development of an immunoassay using only monoclonal antibodies may resolve barriers to commercialization.

The availability of sequence data for amylase isoforms in wheat genomes from multiple cultivars has provided the opportunity to analyze these sequences to identify regions (amino acid sequences) that are conserved and others that are variable between isoforms [19,20,30,31]. As a result, it is now theoretically possible to design antigen peptides to generate monoclonal antibodies (produced by immortal cell lines (hybridomas)) that specifically recognize a single α-amylase isoform or alternatively have broad specificity to a variety of isoforms. In combination with current State-of-the-Art detection procedures, these tools would allow development of “next generation” immunoassays capable of sensitively determining amylase levels and differentiating between amylase isoforms, both rapidly and inexpensively [29,32].

Here we report the assessment of expression α-amylase isoforms during wheat grain development and germination, and the subsequent development of three novel peptide monoclonal antibodies designed to target either single or multiple wheat α-amylase isoforms. We assess their specificity and demonstrate their utility in Western blotting and immuno-pulldowns, to demonstrate their potential utility in developing enzyme-linked immunosorbant assays (ELISA) for the wheat industry.

## 2. Results

### 2.1. Expression of Analysis of Wheat (Triticum astevium; TA) α-Amylase (AMY) Genes 

To determine which proteins were the best targets for antibody development and to enable us to design an inducible system for antibody assessment, the expression of wheat α-amylase transcripts during grain development and germination was examined. To examine expression under “normal” conditions, experiments were performed in the LMA resistant, tall *Rht-1* wild-type line Chinese Spring, using primer sequences derived from the Chinese Spring sequence designed in this and in other studies (Appendix A; [19]).

Quantitative reverse transcription-PCR (qRT-PCR) was used to evaluate *TaAMY* gene expression in Chinese Spring during seed development. Based on previous studies, it was predicted that transcripts for *TaAMY1*, *TaAMY2*, *TaAMY3,* and *TaAMY4* would be present during grain development and in mature wheat aleurone tissue [19], and that *TaAMY1* and *TaAMY2* would be dramatically upregulated by GA treatment [20,33]. Transcript analysis of immature aleurones 15–20 days post anthesis (DPA) demonstrated significantly lower levels of *TaAMY1* and *TaAMY2* expression relative to *TaAMY4* starting at 19 DPA, and no change in *TaAMY1* expression across the developmental window assayed (Figure 1a). *TaAMY* gene expression was normalized relative to *TaAMY1* at 15 DPA. The expression of *TaAMY2* was only significantly elevated at 19 DPA, when it was expressed two-fold higher than *TaAMY1* (*p*-value ≤ 0.05; Figure 1b). As with *TaAMY1* and *TaAMY2*, *TaAMY4* expression levels increased 4-fold at 19 DPA (*p*-value < 0.05; Figure 1d).

Analysis of *TaAMY* expression in immature whole seeds from 15 to 25 DPA was consistent with what was observed in immature aleurones, with similar expression levels of *TaAMY1*, *TaAMY2*, and *TaAMY4* through 19 DPA (Figure 2). There was no significant change in *TaAMY1* expression across the observed time course (Figure 2a). As before, expression levels of *TaAMY2* were elevated at 19 DPA, being expressed two-fold higher than *TaAMY1* (*p*-value ≤ 0.05) and remained elevated through to 21 DPA (Figure 2b). At 19 DPA, *TaAMY4* levels increased 4-fold and continued to rise until reaching peak levels at 23 DPA; being more than 400-fold higher than either *TaAMY1* or *TaAMY2* (*p*-value ≤ 0.0001; Figure 2d).

In addition, expression of *TaAMY1*, *TaAMY*2, and *TaAMY4* was compared in mature aleurones imbibed without or with GA (Figure 3). *G*ene expression was normalized relative to *TaAMY1* without GA treatment. In the absence of GA, expression of *TaAMY4* was 141-fold higher than *TaAMY1* and 55-fold higher than *TaAMY2* (*p*-value ≤ 0.0001). *TaAMY2* expression was also 2.5-fold higher than *TaAMY1* (*p*-value ≤ 0.05). The presence of GA resulted in a 318-fold increase in *TaAMY1* (*p*-value ≤ 0.0001) and a 1124-fold increase in *TaAMY2* (*p*-value ≤ 0.0001). However, the presence of GA resulted in a 13-fold decrease in *TaAMY4* (*p*-value ≤ 0.0001). No expression of *TaAMY3* was detected in any of the samples evaluated (Figure 1c, Figure 2c, and Figure 3).

### 2.2. Monoclonal Antibody Development

Three novel monoclonal peptide antibodies were developed for detecting elevated levels of α-amylase in wheat grain. Peptides were selected based on the predicted TaAMY1 protein sequence which is known to be expressed during grain development, LMA, and germination (Figure 4; [20,30]). The decision to develop monoclonal antibodies to short peptide sequences, rather than full length proteins, was made with the goal of generating antibodies with different specificities and utilities. The four wheat α-amylase isozymes TaAMY1, TaAMY2, TaAMY3, and TaAMY4 act at different times in development and contribute to LMA and PHS in distinct ways. Though the proteins share a high degree of amino acid sequence homology overall (64–75%), some regions share very high levels of amino acid identity while others are relatively divergent [19]. Epitopes recognized by antibodies are typically 5–6 amino acids long. Consequently, at least theoretically, peptides can be designed to isoform-specific regions of these proteins to produce isoform-specific antibodies; or alternatively, by targeting highly conserved regions (found in many isoforms), antibodies that detect multiple isoforms can be generated. With the goal of developing both specific and generalist antibodies to wheat α-amylases, three peptide sequences were used in this study (Figure 4). The peptide AMY1-A was used to generate a novel monoclonal antibody predicted to be specific for TaAMY1 protein. Peptide AMY1-B was used to generate a monoclonal antibody predicted to recognize either TaAMY1, or both TaAMY1 and TaAMY2, and contained partial amino acid overlap with polyclonal antibodies previously developed [28]. Peptide AMY1-C was used to generate a novel generalist monoclonal antibody predicted to recognize TaAMYs 1, 2 and 4.

### 2.3. Validation of TaAMY Antibodies for α-Amylase Detection

Alpha-amylase enzymes are released from the aleurone in response to GA signaling during germination and digest starch in the endosperm to mobilize stored energy reserves and as shown above they are highly induced by GA treatment. Therefore, it was hypothesized that if the TaAMY antibodies developed were target specific then they should detect α-amylase proteins in a GA-dependent fashion, i.e., dramatically stronger positive signals are expected in GA-treated samples. Initial immunoblot analysis was performed using anti-TaAMY1-A at a dilution of 1:100, anti-TaAMY1-B at a 1:100 dilution, and anti-TaAMY1-C at a dilution of 1:500, to detect α-amylase proteins in aleurone secretions and ground aleurone tissues imbibed in the absence or presence of 10 μM GA_3_ (Figure 5a). The predicted molecular weights of the four wheat α-amylase isozymes range from 45.37 to 48.32 kDa [19]. All three antibodies cross-reacted with a single protein band migrating at approximately 48 kDa in the secreted fractions from GA-treated aleurones. The same band was detected in GA-treated ground aleurone tissues with anti-TaAMY1-B and anti-TaAMY1-C antibodies. Differences in antibody avidity (i.e., affinity for the protein target), antibody specificity, and/or overall endogenous protein expression levels may account for the differences in amylase detection in GA-treated aleurone tissues. To confirm that the protein bands recognized by anti-TaAMY1-C contained wheat α-amylase, peptide sequencing was performed on the proteins corresponding to the immuno-positive band from the Chinese Spring GA-secreted fractions. This analysis identified amino acid peptide sequences that aligned to α-amylase isoforms TaAMY1, TaAMY2, and TaAMY4. In total, 30 unique peptides were identified across the three TaAmy1 homoeoforms (Appendix A). Percent coverages for TaAMY1.1, TaAMY1.2, and TaAmy1.3 were 45.33%, 43.79%, and 45.9%, respectively. A total of 27 unique peptides were identified for a single TaAMY2 homoeoform and represented 55% amino acid sequence coverage (Appendix A). Additionally, a total of 22 unique peptides were identified across two TaAMY4 homoeoforms, with percent coverages of 41.5% (TaAMY4.1), and 39.6% (TaAMY4.2, Appendix A). Since the peptide sequences used to generate antibodies were derived from the Chinese Spring genome sequence, whether or not the antibodies recognized α-amylase in a range of wheat germplasms were assessed (Figure 5b,c). Immunoblot analysis using anti-TaAMY1-C at a dilution of 1:500 was performed on the secreted aleurone fractions from a broad collection of wheat varieties (Figure 5b,c). Samples differed by hardness (soft vs. hard), color (red vs. white), growing season (winter vs. spring), and degree of cultivation (modern hybrid vs. historic variety). These samples also differed by geographical production area originating from the United States, Mexico, Europe, Australia, and the Middle East. As with Chinese Spring, anti-TaAMY1-C detected a putative α-amylase band running at approximately 48 kDa in the secreted fractions from the GA-treated aleurones that was absent in the non-GA treatment, and this band was identified in all wheat cultivars tested. This result is consistent with previous studies that reported a high-degree of nucleotide and amino acid sequence similarity between α-amylase homoeoforms within, and across, cultivars [19,34,35]. This result suggests that anti-TaAMY1-C will be suitable for detecting α-amylase across diverse wheat germplasms.

### 2.4. Validation of TaAMY Antibody Specificity

The α-amylase peptide sequences used for monoclonal antibody design, were selected based on their being found only in TaAMY1 (as in anti-TaAMY1-A) or being highly conserved across three of the four α-amylase isozymes (as in anti-TaAMY1-C). To assess whether the antibodies possessed their intended specificities required us to have protein samples known to possess a single TaAMY isoform. To this end, separate lines of *E. coli* were generated that heterologously expressed individual 6xHis-tagged isoforms. The performance of anti-TaAMY1-A antibody as a specific antibody, and anti-TaAMY1-C antibody as a generalist antibody were then evaluated using an immunoblot analysis of the heterologously expressed 6xHis-tagged recombinant proteins from *TaAMY1*, *TaAMY2*, and *TaAMY4* genes (Figure 6a). Secreted protein fractions from untreated and GA-treated aleurones, and a protein sample from an *E. coli* line transformed only with the empty expression vector were used as experimental controls. Additionally, immunoblot analysis with an anti-6xHIS antibody was performed to confirm the expression and identity of the recombinant 6xHis-tagged proteins.

Both anti-TaAMY-1A and anti-TaAMY1-C detected the 48 kDa α-amylase protein band in the GA-treated, but not the untreated, aleurone secretion fraction. The anti-TaAMY1-A antibody also specifically detected HIS-tagged TaAMY1 protein but not HIS-tagged TaAMY2 or HIS-tagged TaAMY4. In contrast, anti-TaAMY1-C detected all three HIS-tagged proteins. This result was consistent with the fact that the peptide sequence used to generate anti-TaAMY1-A is unique to TaAMY1, whereas the sequence used to generate anti-TaAMY1-C is conserved across the wheat α-amylase isoforms (Figure 4). The bands recognized by anti-TaAMY antibodies also cross-reacted with anti-HIS antibody and subsequent peptide sequencing of the immune-positive proteins confirmed that the sequences of the most abundant peptides recovered from these samples aligned with the amino acid sequences of HIS-tagged TaAMY1, 2, and 4 proteins in their respective expression lines (Appendix A).

### 2.5. Evaluating the Use TaAMY Antibodies in Pairs 

Immunoassays, including lateral flow immunoassays (LFIs) and enzyme-linked immunosorbent assays (ELISAs), use two distinct antibodies to capture and subsequently detect specific protein targets [2]. The antibodies developed in this study were evaluated for their potential to detect α-amylase in pairs in both standard immuno-pulldown assays (employing enzyme-linked anti-mouse secondary antibodies) (Figure 6b), and immuno-pulldown using a gold-conjugated detection antibody (Figure 6c,d). In both cases all three antibodies were evaluated for use as the capture antibody, while anti-TaAMY1-C was evaluated for use as the detection antibody. 

In standard immuno-pulldowns, all three TaAMY antibodies were compatible for use as the capture antibody in combination with the anti-TaAMY1-C detection antibody. Immuno-pulldowns resulted in the detection of the 48 kDa α-amylase target (Figure 6b). In addition to detecting α-amylase, two additional bands migrating at ~25 kDa and 50 kDa were also detected in all cases. Interestingly, the signal intensity of these two bands was much lower when anti-TaAMY1-A was used for target capture. The additional bands detected were hypothesized to represent non-specific detection of the heavy- and light-chains of anti-Ta-AMY1-C by the anti-mouse alkaline phosphatase secondary antibody used in immunoblot analysis. To test this hypothesis, and to evaluate the potential for use in LFI or ELISA platforms, an alternative method of signal detection was employed to avoid the need to use the anti-mouse alkaline phosphatase secondary antibody. To achieve this anti-TaAMY1-C was conjugated to gold particles to allow direct visual detection of this antibody (Figure 6c,d). The specificity and signal strength of gold-conjugated anti-TaAMY1-C was then evaluated when used alone, and in combination with each of three TaAMY capture antibodies.

Immunoblots representing three dilutions of soluble secreted fractions from GA-treated aleurones were processed using gold-conjugated anti-TaAMY1-C as the detection antibody (Figure 6c). As before, the 48 kDa α-amylase band was detected, and the signal intensity associated with gold conjugated anti-TaAMY1-C binding was consistent with the quantity of protein target loaded, i.e., the lane with the highest color intensity corresponded with that with the highest protein loading. Next, immuno-pull downs were performed, as before, to assess the effectiveness of each of the three TaAMY capture antibodies paired with gold conjugated anti-TaAMY1-C for detection. Again, gold-conjugated anti-TaAMY1-C detected the 48 kDa α-amylase band regardless of which antibody was used for protein capture (Figure 6d). Importantly the ~25 and 50 kDa bands previously detected were absent, which is consistent with our hypothesis that these bands represent detection of the heavy- and light-chains of anti-TaAMY1-C by the anti-mouse alkaline phosphatase secondary antibody. These results suggest that the novel monoclonal antibodies described in this study are suitable for use in LFI and ELISA platforms for the detection of wheat α-amylase targets.

## 3. Discussion

### 3.1. Overview

Wheat α-amylases are essential for starch digestion in seed germination [10]. However, expression of these enzymes at the wrong time during grain development (LMA), or in mature seeds prior to harvest in response to rain (PHS), can negatively impact flour end-use quality. Increasing temperatures may increase the risk of LMA and PHS [36,37]. Increasing climate variability appears to be associated with more frequent Falling Number problems in wheat [37,38,39,40,41,42,43]. To prevent wheat end-use quality problems due to α-amylase, it is essential to have more accessible, rapid, and isoform-specific tests than the Falling Number method for determining if grain samples are sound, and if not, the degree of loss in quality caused. With this in mind, this study evaluated the characteristics of three novel monoclonal antibodies to wheat α-amylase(s) to assess their potential for future use in diagnostic immuno-assays for LMA and PHS. The long-term goals of this work are to help to mitigate economic losses and provide research tools to help unravel the genetic causes of these distinct phenomena to facilitate targeted breeding for resistance.

### 3.2. Expression Analysis of Wheat α-Amylase Genes

Transcript analysis of α-amylase genes in Chinese Spring wheat, was used to confirm the identities of the isoforms involved and identify optimal timepoints for cloning. Our results confirmed previous findings that *TaAMY1*, *TaAMY2*, and *TaAMY4* are differentially expressed across development, and in response to GA treatment in mature seeds [19,20]. Furthermore, both *TaAMY1* and *TaAMY2* were highly induced with GA in mature aleurones consistent with their previously reported roles in seed germination and PHS [17,19,20]. Across seed development, expression levels of both *TaAMY2* and *TaAMY4* increased at 19 DPA, whereas *TaAMY1* did not. Overall levels of *TaAMY4* were significantly higher than both *TaAMY1* and *TaAMY2* during development, peaking at around 23 DPA. *TaAMY4* levels were also detected in mature imbibing aleurones, with or without GA treatment, although these levels, unlike *TaAMY1* and *TaAMY2*, were not upregulated by GA.

Examination of *TaAMY1*, *TaAMY2*, *TaAMY3*, and *TaAMY4* mRNA expressions during Chinese Spring grain development and in response to the germination-signal GA in imbibing mature aleurones provided some additional insights into the expression of these gene families. Failure to detect the *AMY3* expression led to the conclusion that it is not a major source of grain α-amylase (Figure 1, Figure 2 and Figure 3). Thus, emphasis was placed on raising antibodies that primarily detected the predicted TaAMY1 protein, and also TaAMY2 and TaAMY4 proteins. The GA-insensitive semi-dominant *Reduced height* (*Rht-B1b* and *Rht-D1b*) alleles in wheat are associated with higher tolerance to LMA [44]. Chinese Spring is a known LMA resistant line that carries the wild-type *Rht-B1a* and *Rht-D1a* alleles resulting in a tall growth habit [45]. Thus, LMA resistance in Chinese Spring is independent of the *Rht* genes. Failure to detect *TaAMY3* mRNA during development (17 to 26 DPA) and during GA treatment of germinating mature aleurones was consistent with previous studies that failed to detect *TaAMY3* mRNA in *Rht-1* wild-type LMA resistant and LMA constitutive (susceptible lines that express the trait without cool treatment) from the ‘Maringa’/’Spica’ population [20]. Mieog et al., [19] detected very low levels of *TaAMY3* transcript during development that decreased during germination using one of the two primer sets used in the current study. Given that this experiment was performed using a combination of lines enriched for the *Rht-1* wild-type alleles in an effort to enrich for LMA constitutive lines, it is possible that the *TaAMY3* expression is increased by a different genetic source of LMA susceptibility than was used in the Barrero et al., [20] study. Future work could examine this possibility by measuring expression in single wheat lines from the Mieog et al., [19] study. *TaAMY1*, *TaAMY2*, and *TaAMY4* were all expressed during seed development, and *TaAMY4* showed a significant increase in expression levels between 20 and 23 DPA in LMA resistant Chinese Spring. This observation was unique to this study. Future work should examine whether *TaAMY4* can be associated with LMA in LMA susceptible backgrounds. As previously reported, both *TaAMY1* and *TaAMY2* showed strong GA-induction in mature aleurones. Interestingly, *TaAMY4* mRNA levels actually decreased with GA treatment. This suggests that *TaAMY1* and *TaAMY2* are the major GA-induced α-amylase transcripts in germinating mature aleurones in Chinese Spring (Figure 3). Previous work suggested that LMA (also called pre-maturity α-amylase or PMA) is induced by GA hormone and that LMA-inducing cool temperatures increase kernel GA sensitivity [20,46,47,48,49]. This observation, however, may depend on the genetic source of LMA susceptibility, because it appears that the *LMA-1* susceptibility gene on wheat chromosome 7B does not function by increasing GA hormone levels [50]. Future work will need to better examine whether genes controlling LMA susceptibility do so via different mechanisms.

### 3.3. Potential for an α-Amylase Immunoassay for Wheat

The data presented here demonstrate that it is feasible to detect wheat α-amylases using a combination of two monoclonal antibodies, one to capture and one to detect α-amylase (Figure 6b). This is an improvement upon using a monoclonal and polyclonal antibody combination because monoclonals are a renewable reagent using immortalized cell lines. The anti-TaAMY1-C antibody was chosen as the detection antibody because it appeared to have stronger avidity in preliminary experiments when conjugated to gold particles. Moreover, anti-TaAMY1-C was able to recognize α-amylase in a wide range of germplasms (Figure 5b,c). The feasibility of this approach raises the possibility of developing either a lateral flow immunoassay (LFIs) or enzyme-linked immunosorbent assays (ELISAs) for wheat α-amylase. Currently, the wheat industry relies on the Falling Number method to detect α-amylase from preharvest sprouting and LMA as a risk to end-product quality. The Falling Number is influenced not only by α-amylase, but by grain protein and starch composition as well (reviewed by [2]). Immunoassays represent an improvement in that they are highly specific, require less expensive equipment, and subject to lower risk of user error.

## 4. Materials and Methods 

### 4.1. Plant Material and Growth Conditions

*Triticum aestivum* Chinese Spring plants were grown in a glasshouse with supplemented natural light (using a sodium lamp) to achieve 400 μmol quanta m^−2^ s^−1^, and a 16 h day/8 h night photoperiod at the WSU Plant Growth Facility as in [51]. Day temperatures ranged from 21 to 24 °C, and night temperatures ranged from 15 to 18 °C. Relative humidity ranged from 40 to 75%. Seeds were harvested from spikes at 15–25 days past anthesis (DPA), and at harvest maturity. Samples harvested before harvest maturity (15–25 DPA) were flash frozen with liquid nitrogen and stored at −80 °C until all samples were processed. Mature spikes (42–45 DPA) were collected, hand threshed, and the seeds stored in individual envelopes at room temperature. In addition to Chinese Spring (hard red spring; HRS), alpha-amylase expression was examined from a broad collection of wheat across market classes including: Pacific Northwest soft white winter wheats (SWW) ‘Jasper’ (CV-1124, PI 678442 [52]), ’Otto’ (CV-1087, PI 667557 [53]) ‘SY Ovation’ (PVP 201100387), and club wheat ’Bruehl’ (PI 606764 [54]), soft white spring wheat (SWS), ‘WA8124′ (IDO599/S2K00095, [51], soft white spring club wheat ‘JD’ (PI 63098), Australian HWSs ‘Seri 82′ (PI591774 [55]), ‘Spica’ (PI213830 [20,50] and ‘Halberd’(PI377885 [56]). This material was obtained either from the Washington State University Cereal variety trials (Cereal Variety testing 2020), or grown at Spillman Agronomy Farm (Pullman, WA, USA) in 2020. Additionally, seed samples for the landrace HWS wheat, Eden, the landrace soft red winter wheat, English Squarehead, and the landrace SWS, Sonora, were obtained from Palouse Heritage (www.palouseheritage.com), and grown at Palouse Colony Farm (Endicott, WA, USA) in 2020. 

### 4.2. Expression Analysis of α-Amylase Genes in Grain Development

*TaAMY* expression analysis was performed using Chinese Spring across a developmental time course from 15 to 20 days post anthesis (DPA) with immature aleurones, 15–25 DPA with immature seeds, and mature aleurone layers without or with 10 µM GA_3_ treatment. Immature aleurone samples were removed directly from immature seeds and harvested as described above. All samples were flash frozen with liquid nitrogen and ground in a mortar and pestle. Total RNA was extracted using a phenol–chloroform extraction method adapted for recalcitrant tissues as described in [57,58]. Prior to synthesizing complementary DNA (cDNA), genomic DNA contamination was removed using the DNA-*free*™ DNA Removal Kit according to the manufacturer’s protocol (Thermo Fisher Scientific Inc.; Waltham, MA, USA). cDNA was then synthesized using 100 ng of total RNA with the ProtoScript^®^ First Strand cDNA Synthesis Kit (New England Biolabs, Ipswich, MA, USA).

All reverse transcriptase real-time quantitative-PCR (RT-qPCR) experiments were performed using gene specific primers on a BioRad CFX96™ Real-Time PCR thermocycler ([19,59,60]; Appendix A). For each gene, expression analysis was performed across three biological replicates using SsoAdvanced™ Universal Sybr^®^ Green Supermix (Bio-Rad, Hercules, CA, USA). For each reaction, the starting cDNA template was diluted 1:20, and the starting concentration for each primer was 1 μM. Cycling conditions consisted at 95 °C for 30 s, then 40 cycles at 95 °C for 25 s followed by 62 °C for 20 s. Transcript levels were normalized against the *TaActin* housekeeping control [19]. For each experiment, the relative expression of *TaAMY* genes was calculated using the Delta-Delta CT method normalized to *TaAMY1* expression in the untreated control [61].

### 4.3. Cloning of α-Amylase Genes and Recombinant Protein Expression

Gene specific primers were developed from consensus coding sequences of *Triticum aestivum* ‘Chara’ ([19]; Appendix A and aligned using Geneious Prime 2019.2.1 software (https://www.geneious.com). Coding sequences of *T. aestivum α-amylase 1*, *2*, and *4* (*TaAMY1*, *TaAMY2*, *TaAMY4)* were amplified from the normalized Chinese Spring cDNA templates generated for expression analysis, by polymerase chain reaction (PCR) using KOD Hot Start DNA Polymerase proofreading Taq polymerase (Novagen, Pasig, Philippines). PCR cycling conditions were: 2 min denaturation at 95 °C, followed by 35 cycles of 25 s at 95 °C, 20 s at 60 °C (*TaAMY1*), or 63 °C (*TaAMY2)*, or 59 °C (*TaAMY4*), and for 25 s at 70 °C. An A residue was added to each blunt-ended amplicon using Taq DNA Polymerase (New England Biolabs, Ipswich, MA, USA) according to the manufacturer’s protocol to facilitate TA cloning. Amplicons were gel purified using a QIAEX Gel Extraction Kit (Qiagen, Germantown, MD, USA), ligated into TOPO TA cloning vector pCR 2.1 (Invitrogen, Carlsbad, CA, USA), and transformed into One Shot TOP10 cells (ThermoFisher, Waltham, MA, USA). PCR colony screens were performed to confirm amplicon insertion, and the plasmid inserts were sequenced by Eurofins Genomics (Louisville, KY, USA). 

For protein expression, *TaAMY* sequences were amplified from the cloning vector using primers engineered to add restrictions sites (*Eco*RI-HF and *Bgl*II) to flank the coding region, to facilitate in-frame insertion into the expression vector pRSET-C (ThermoFisher Scientific, Waltham, MA, USA; Appendix A). PCR cycling conditions were the same as those listed for cDNA amplification. Amplified inserts and pRSET-C were digested with *Eco*RI-HF and *Bgl*II) at 37 °C for 2 h, separated by agarose gel electrophoresis and gel purified. Insert and vector fragments were ligated at 4 °C for 12 h using T4 ligase (BioRad, Hercules, CA). Ligations were transformed into, and maintained in, DH5- α chemically competent *E. coli* with 75 µg/mL carbenicillin selection (75 µg/mL). Colony screens were performed to confirm the presence of the inserts and sequenced (Eurofins Genomics, Louisville, KY, USA) to confirm in-frame insertion into pRSET-C. 

For protein expression, *E. coli* strain BL21 Star was transformed separately with expression vectors carrying *TaAMY1*, *TaAMY2* or *TaAMY4* and the resulting cell lines carrying each plasmid were amplified by growing in 5 mL LB liquid media (MP Biomedicals) at 37 °C with shaking at 150 rpm overnight. After incubation, 15 mL of pre-warmed overexpression broth (OB; Dual Media Set, ZymoResearch, Irvine, CA, USA) was added and cultures incubated for up to 4 h. Aliquots of 2 mL were harvested from each culture, spun down, and the pellets frozen at −20 °C for 24 h. Frozen pellets were lysed using the BugBuster Plus Lysonase Kit and resuspended in BugBuster Protein Extraction Reagent according to the manufacturer’s instructions (Millipore Sigma, St Louis, MO, USA). Crude protein lysates were then centrifuged at 4 °C for 20 min at 12,000 rpm. The resulting supernatants were transferred to clean microfuge tubes, and the insoluble lysate pellets resuspended in 300 µL PBS-T. Aliquots of 40 µL from all samples were mixed with an equal volume of 2× Laemmli buffer and then denatured at 95 °C for 10 min. For monoclonal antibody validation, protein samples were separated on triplicate SDS-PAGE gels, two of which were blotted to PVDF, and one Coomassie stained. The blotted gels (on PVDF membrane) were incubated with either anti-TaAMY1-A antibody at a dilution of 1:100, or anti-TaAMY1-C antibody at a dilution of 1:500, and mouse anti-6xHIS monoclonal antibody at a concentration of 1:3000 (Thermo Fisher Scientific, Waltham, MA, USA; R932-25). Proteins detected by either anti-TaAMY1-A, or anti-TaAMY1-C; anti-6xHIS antibodies were identified on the PVDF membranes, and gel fragments corresponding to these bands were excised from the Coomassie-stained duplicate gel. These protein-containing gel fragments were then dehydrated with 100% acetonitrile, and shipped to Bioproximity (Manasas, VA, USA) for tryptic digestion and peptide sequencing. 

### 4.4. Statistical Analysis

Analysis of variance (ANOVA) was used to identify statistically significant differences in *TaAMY* gene expression levels in Chinese Spring (Figure 1, Figure 2 and Figure 3). ANOVAs were performed with the PROC GLM function and using a Tukey’s all pairwise comparison performed in SAS version 9.4 (SAS Institute, Cary, NC, USA). For all experiments *p*-values ≤ 0.05 were considered significant.

### 4.5. Peptide Monoclonal Antibody Development

Nucleotide sequences for *Triticum aestivum α-amylase 1* (*TaAMY1*) were downloaded from the National Center for Biotechnology Information (NCBI) and International Wheat Genome Sequencing Consortium (IWGSC) databases [31]. The sequences were translated *in silico* and the resulting amino acid sequences aligned using Clustal W2 software version 2.1 [62]. Allelic polymorphism between the multiple copies of each α-amylase gene within the Chinese Spring reference genome and between sequences available for different cultivars was assessed to identify invariant regions between gene copies, and across genotypes, to ensure that peptide sequences chosen for antibody production were of broad utility. Amino acid sequences of the other α-amylase isoforms (*TaAMY2*, *3,* and *4*) were downloaded from the NCBI database as FASTA text files, aligned to determine regions of highest sequence homology and divergence, and then analyzed using the OptimumAntigen^TM^ design tool (GenScript) to identify regions predicted to be both highly antigenic and hydrophilic. After combining the results of these analyses, three 14 amino acid peptides of *TaAMY1* were designed for monoclonal antibody production. Peptide synthesis and antibody production was outsourced to GenScript Biotech (Piscataway, NJ, USA) Briefly, synthesized peptides were conjugated to keyhole limpet hemocyanin and the conjugates used to immunize BALB/c mice. After two rounds of immunization the antibody producing spleens were harvested from the mice and used to create immortal antibody producing cell lines (hybridomas). Culture supernatants from individual hybridoma cell lines were screened using enzyme-linked immunosorbent assays (ELISAs) against their respective peptide antigens to identify those producing antibodies with highest specificity and cross-reactivity. 

### 4.6. Induction of α-Amylase Enzymes from Aleurone Tissue

Because peptides were selected based on the Chinese spring reference sequence, it was important to determine if the resulting antibodies could detect α-amylase in diverse wheat cultivars of multiple market classes. Mature seeds were surface sterilized with 10% bleach, 0.01% sodium dodecyl sulfate (SDS), the brush and embryo-ends excised, and then the “half” seeds imbibed on 5 mM 2-(N-morpholino) ethane sulfonic acid buffer (MES, pH 5.5) at 22 °C for 2 days in the dark [63]. The starchy endosperm was then removed using a scalpel, and the isolated seed coat/aleurone layers (hereafter referred to as aleurones) was incubated at room temperature with shaking at 50 rpm for 3 days in a 20 mM CaCl_2_, 5 mM MES buffer (pH 5.5) in the presence or absence of 10 µM gibberellin A_3_ (GA_3_) [17]. Aleurones were imbibed at a ratio of 10 aleurone layers to 1.5 mL incubation buffer in a 50 mL conical tube [17]. Soluble secreted aleurone fractions were decanted into 2 mL microfuge tubes and stored for up to 4 weeks at 4 °C. Aleurone layers used for RNA extraction were flash frozen with liquid nitrogen and placed at −80 °C for long term storage. Aleurone layers used for monoclonal antibody validation were flash frozen with liquid nitrogen, ground into a fine powder, placed into 2 mL low-bind plastic microfuge tubes and mixed with 1 mL Phadebas extraction buffer (100 mM sodium malate, 5 mM CaCl_2_, pH 6.0), and incubated for 10 min at 50 °C. They were then centrifuged at 2800 rpm for 10 min at 22 °C. The supernatants were transferred to clean tubes and stored at 4 °C as described above. For all sample types, α-amylase enzyme activity was measured in secreted aleurone fractions and extracted aleurone layers using a high throughput Phadebas method [17,26,51].

### 4.7. Monoclonal Antibody Validation and Peptide Sequencing

For immunoblot analysis, protein concentrations from soluble secreted fractions and whole ground aleurone tissue were determined using the Quickstart Bradford assay (BioRad, Hercules, CA). A total of 20 µg from each sample was mixed with 2× Laemmli SDS-PAGE buffer containing 2.5% 2-ß-mercaptoethanol. Samples were denatured at 95 °C for 10 min and then fractionated on precast “Any kD” TGX gels (BioRad, Hercules, CA, USA) at 100 V for 1 h. Gels were run in duplicate. One was blotted to 0.2 µm polyvinylidene fluoride (PVDF) membranes using a semi-dry Trans-Blot Turbo Transfer System at 25 V for 14 min (BioRad) for immunostaining, the other was stained with Coomassie Blue for peptide sequencing. Starting concentrations of monoclonal antibodies, anti-TaAMY1-A, anti-TaAMY1-B, and anti-TaAMY1-C were 2.14 mg/mL, 1.415 mg/mL, and 1.19 mg/mL, respectively. The antibodies were diluted 1:100, 1:250, and 1:500 in 1× phosphate buffered saline with tween-20 buffer (PBS-T; pH 7.4). Immunoblots were developed using the Western Breeze anti-mouse alkaline phosphatase detection system (BioRad, Hercules, CA, USA) according to the manufacturer’s protocol with exception to the addition of Fish Gelatin (Biotium, Fremont, CA, USA) to a final concentration of 1× in the membrane blocking solution. Immunoblots were imaged using a ChemiDocMP (BioRad, Hercules, CA, USA). Coomassie (Brilliant Blue R 250) stained gels were incubated in destaining solution (50% water, 40% methanol, 10% acetic acid) for 4 h. Gel slices from both non-treated and GA-treated samples corresponding to bands detected by immunoblot analysis were excised, dehydrated with 100% acetonitrile, and shipped to Bioproximity (Manasas, VA, USA) for in-gel tryptic digestion followed by label-free quantitative (LFQ) peptide sequencing using UPLC-MS/MS.

### 4.8. Alpha-Amylase Immunoprecipitation Assays

To determine if antibodies were suitable for use in pairs for the development of lateral flow assays (LFAs) or enzyme-linked immunosorbent assays (ELISAs), immuno- pulldown assays were performed. For each pulldown, 83 µL of Protein A Dynabeads (Invitrogen, Carlsbad, CA, USA) were pipetted into a microfuge tube. The beads were immobilized using a magnetic rack to facilitate removal of the supernatant, and then resuspended in 200 µL of PBS-T buffer (pH 7.4) containing 8 µg of each anti-Ta-AMY antibody. Dynabeads were then incubated with rotation at 100 rpm for 10 min at room temperature, placed in a magnetic rack to remove the supernatant, and washed once with 200 µL of PBS-T. Next, antibody-coated beads were mixed with 200 µL of soluble secreted proteins from GA-treated and non-treated aleurones. As before, samples were incubated with rotation for 10 min at room temperature. After incubation, samples were moved to the magnetic rack, the were supernatants removed, then the beads were resuspended in 100 µL of PBS-T and moved to a clean sample tube. The supernatant was removed once more, and the beads resuspended in 20 µL of 1× PBS-T buffer and mixed with 20 µL 2× Laemmli SDS-PAGE buffer. Samples were then denatured at 95 °C for 10 min and placed on the magnetic rack. The denatured protein containing supernatants were removed and moved to clean tubes. An amount of 20 µL of each sample was subjected to SDS-PAGE and detected as described above under immunoblot analysis with a 1:500 dilution of anti-TaAMY1-C.

To eliminate detection of the light- and heavy-chains of the primary antibody used for immunoprecipitation (by the secondary antibody) and to increase sensitivity, gold particles were conjugated to the anti-TaAMY1-C using the Abcam Gold Conjugation Kit (20 nm, 20 OD) according to the manufacturer’s recommendations (Abcam, Cambridge, UK). Briefly, anti-TaAMY1-C was diluted to a starting concentration of 0.25 mg/mL in the diluent provided. Next, 12 µL of diluted antibody along with a gold 20 nm reaction buffer were combined, and 45 µL added to a tube of 20 nm gold particles. Samples were incubated at room temperature for 15 min, then 5 µL of quenching buffer was added and the reaction incubated for 5 min at room temperature. To remove unbound antibodies, samples were diluted 10-fold in 0.1× quenching buffer, and centrifuged at room temperature at 9000× *g* for 20 min. The supernatant was removed, and the gold particle pellet was resuspended in 50 µL of 0.1× quenching buffer. Immunoblot analysis was performed using 20 µg from secreted aleurone fractions as described above. Membranes were blocked for 30 min with Western Breeze blocking solution supplemented with 1× fish gelatin, washed twice with water for 2 min, and then incubated at room temperature in gold conjugated anti-TaAMY1-C antibody at a dilution of 1:500 in PBS-T for up to 4 h to ensure complete antibody binding and color development.

Immunoprecipitation assays using gold conjugated anti-AMY1-C detection antibody were performed as described above with 8 µg of each anti-TaAMY capture antibody affixed to Protein A Magnetic Beads. After immunoblotting, membranes were blocked for 30 min with Western Breeze blocking solution supplemented with 1× fish gelatin, washed twice for 2 min with nanopure water, and then incubated at room temperature in gold conjugated anti-TaAMY1-C antibody at a dilution of 1:500 in PBS-T buffer for 4 h.

## 5. Conclusions

Novel monoclonal antibodies were developed, anti-TaAMY1-A, B, and C, designed to be either isoform specific or recognize multiple α-amylase isoforms. The ability of these antibodies to detect putative α-amylase protein targets in GA-induced mature aleurone and secreted fractions was confirmed by peptide sequencing of the immuno-positive bands. Using *E. coli* lines engineered to express individual 6XHis-tagged isoforms, it was also demonstrated that two of the three monoclonal antibodies developed, anti-TaAMY1-A and anti-TaAMY1-C exhibited the specificity for which they were designed. The anti-TaAMY1-A antibody raised to a sequence specific to TaAMY1 detected only 6XHis:TaAMY1 protein. In contrast, the anti-TaAMY1-C antibody, raised to a peptide sequence predicted to be highly conserved in three of four of the classes of wheat α-amylases, TaAMY1, TaAMY2, and TaAMY4, detected the corresponding three recombinant α-amylase proteins. Furthermore, TaAMY1-C detected the same α-amylase protein targets across diverse wheat market classes, suggesting that it will be of broad utility in the wheat industry. Evaluations of the developed antibodies in pairs, using immuno-pulldown techniques, and in combination with conjugated gold particles, a reagent commonly used in lateral flow immunoassays, demonstrated that all three antibodies could capture their protein targets, and that gold labeled anti-TaAMY1-C is effective in providing the visual detection needed for assay development. In summary, these results confirm that by designing peptide immunogens to more, or less, conserved regions of wheat α-amylases, it is feasible to generate monoclonal antibodies with targeted isoform specificities. The antibodies developed are being evaluated for use in novel rapid testing immuno-assay platforms. Such assays have great potential in helping to overcome limitations of indirect tests like FN.

## Figures and Tables

**Figure 1 plants-12-03798-f001:**
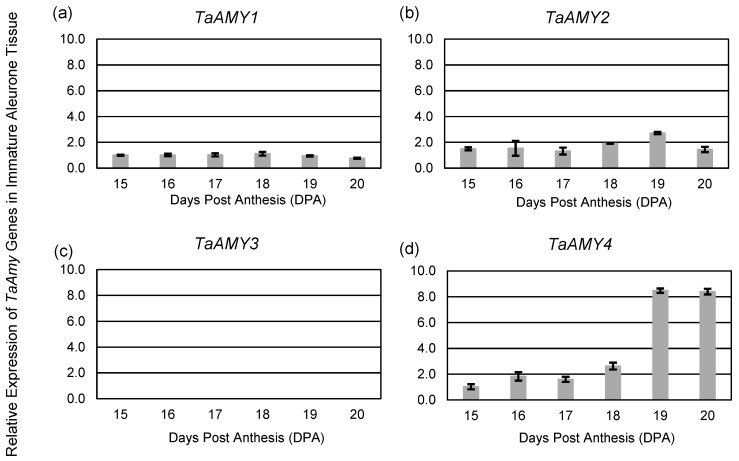
Expression of *TaAmy* transcripts in immature Chinese Spring aleurone tissues. (**a**) *TaAMY1*, (**b**) *TaAMY2*, (**c**) *TaAMY3*, and (**d**) *TaAMY4* gene expressions were evaluated by quantitative RT-PCR analysis in immature Chinese Spring aleurone tissues from 15 to 20 days post anthesis (DPA). *TaAMY* levels are shown relative to the *TaActin* constitutive control: *TaAMY1* time point at 15 DPA is set to 1, allowing comparison of the four transcripts. The mean fold change is shown for three biological replicates and error bars = SD. Statistical significance was determined using an analysis of variance (ANOVA).

**Figure 2 plants-12-03798-f002:**
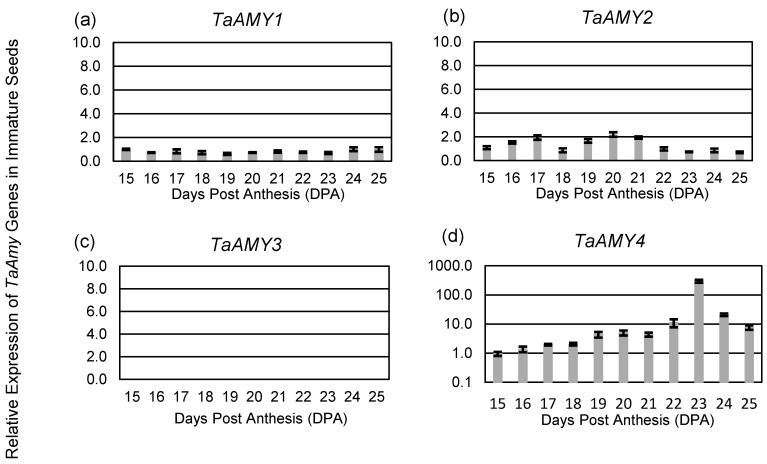
Expression of *TaAMY* transcripts in immature seeds. *TaAMY1* (**a**), *TaAMY2* (**b**), *TaAMY3* (**c**), and *TaAMY4* (**d**) gene expressions were evaluated by quantitative RT-PCR analysis in Chinese Spring immature seeds from 15 to 25 days post anthesis (DPA). *TaAMY* levels are shown relative to the *TaActin* constitutive control: *TaAMY*1 time point at 15 DPA is set to 1. The mean fold change is shown for three biological replicates and error bars = SD. Statistical significance was determined using an ANOVA.

**Figure 3 plants-12-03798-f003:**
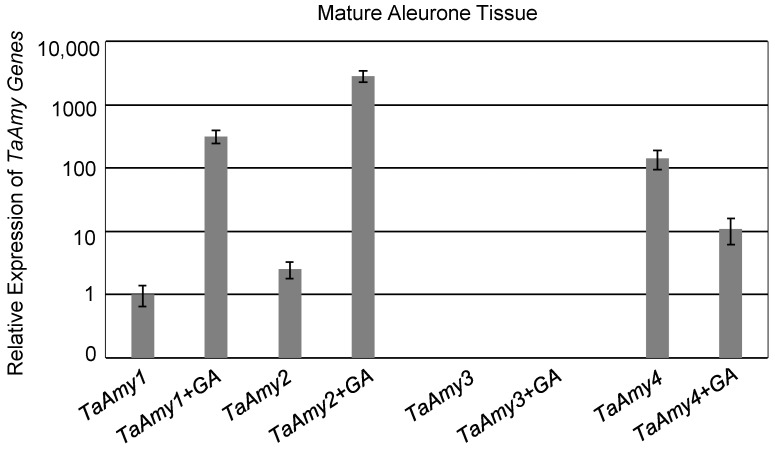
Expression of *TaAmy* transcripts in imbibing mature aleurones. Quantitative RT-PCR analysis of *TaAMY1*, *TaAMY2*, *TaAMY3*, and *TaAMY4* gene expressions were evaluated in imbibing Chinese Spring mature aleurones without and with 10 µM GA_3_. *TaAMY* levels are shown relative to the *TaActin* constitutive control: *TaAmy1* imbibed without GA is set to 1. The mean fold change is shown for three biological replicates and error bars = SD. Statistical significance was determined using an ANOVA.

**Figure 4 plants-12-03798-f004:**
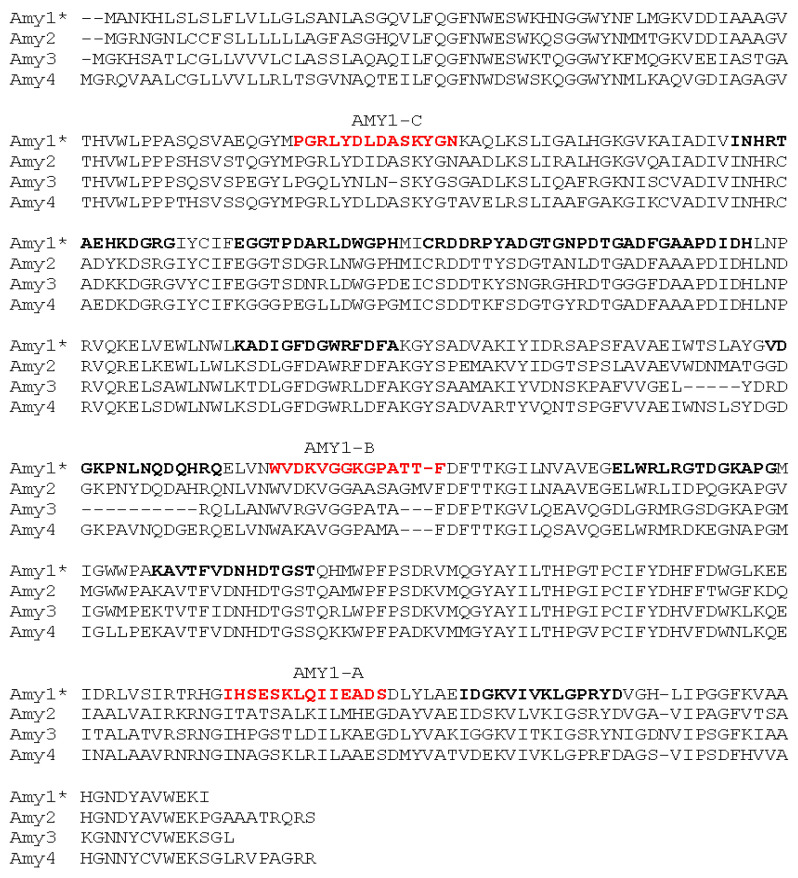
Sequence alignments of four classes of wheat α-amylases used for monoclonal antibody development. BOLD (black and red): Antigenic determinants predicted using the OptimumAntigen^TM^ design tool (GenScript). Analysis was based on antigenicity, hydrophilicity, hydrophobicity, surface probability, transmembrane, homology, flexible region, helix region, sheet region, signal peptide, and modification. RED: Sequences chosen for peptide synthesis and monoclonal antibody production. GenBank Accession numbers: *Amy1*, ATY36099.1. *Amy2,* ATY36104.1. AMY3, X05809. *Amy4,* ATY36106. An asterisk was used to identify the primary peptide sequence.

**Figure 5 plants-12-03798-f005:**
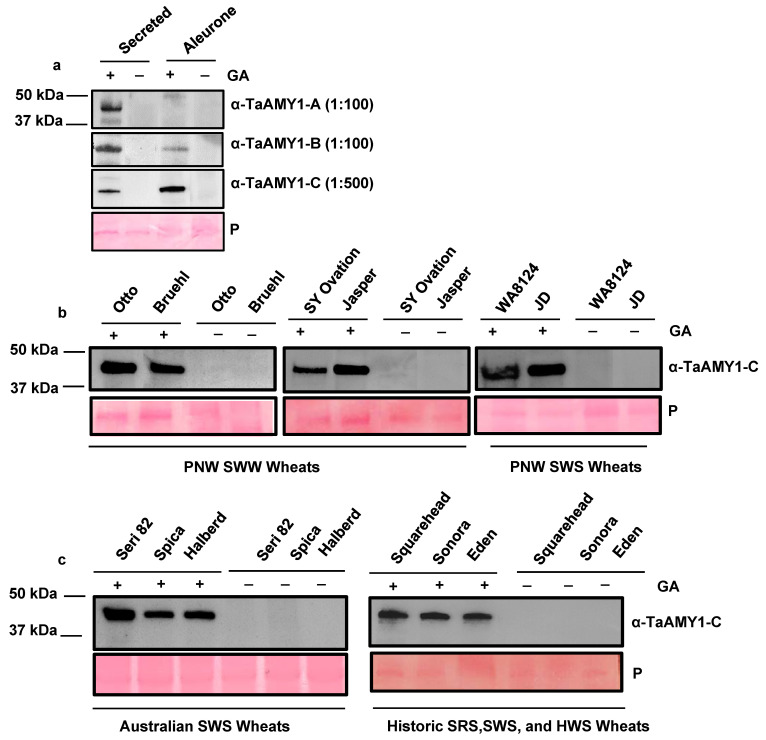
Detection of α-amylase proteins using three novel monoclonal α-amylase peptide antibodies in diverse wheat varieties. Mature aleurone tissues isolated from (**a**) Chinese Spring, (**b**) Pacific Northwest soft white winter (SWW) and soft white spring (SWS) cultivars, and (**c**) Australian SWS wheats (Seri 82, Spica, and Halberd), a landrace soft red winter (SRW; English Squarehead), a landrace SWS (Sonora), and a landrace hard white spring (HWS; Eden) wheat. All samples (aleurone layers) were incubated with shaking for 3 days at room temperature in 10 mM CaCl_2_ buffer with or without 10 µM GA_3_. A total of 20 µg of protein from the secreted aleurone fractions was loaded per lane and detected with (**a**) TaAMY1-A, B (1:100), and C (1:500), and (**b**) TaAMY1-C (1:500) antibodies. A Ponceau stained blot (P) was included with all immunoblots to show protein loading.

**Figure 6 plants-12-03798-f006:**
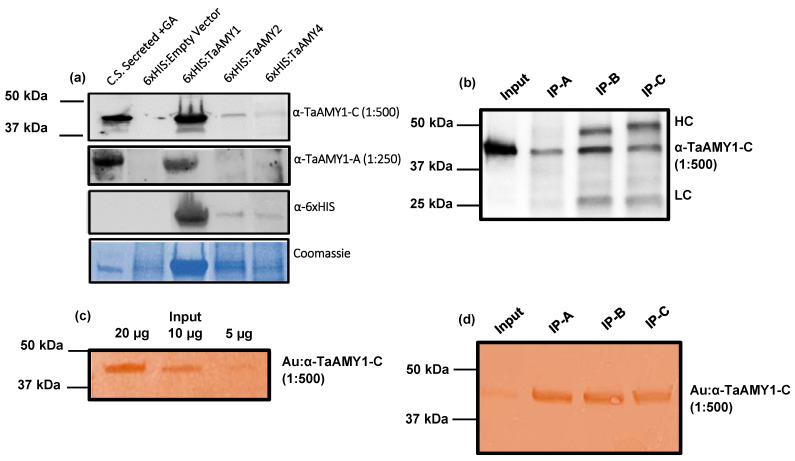
Detection of heterologously (in *E. coli*) expressed TaAMY proteins and immunoprecipitation of endogenous α-amylase proteins using TaAMY1-A, TaAMY1-B, and TaAMY1-C monoclonal antibodies. (**a**) An immunoprecipitation (IP) pipeline was created to validate if the WSU–developed α-amylase antibodies could be used in sandwich ELISAs. (**b**) TaAMY1-A, B, and C antibodies were bound to Protein A Magnetic Beads (Invitrogen) and used to IP α-amylase targets from GA-induced aleurone secretions. Immunoblots were visualized using TaAMY1-C antibody (GenScript 1:500) and (**c**,**d**) Western blots of IP fractions were detected using gold conjugated TaAMY1-C antibody (AbCam). TaAMY1-A, B, and C were all capable of capturing α-amylase protein targets, and gold-linked TaAMY1-C was able to detect α-amylase.

## Data Availability

The data presented in this study are available in the Appendix A listed above.

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
