# Peer review of "Development of Novel Monoclonal Antibodies to Wheat Alpha-Amylases Associated with Grain Quality Problems That Are Increasing with Climate Change"

_plants, 2023, doi:10.3390/plants12223798_

Round 1
Reviewer 1 Report
Comments and Suggestions for Authors
This is a review for the manuscript titled “Development of novel monoclonal antibodies to wheat Al-Pha-Amylases associated with gran quality problems that are increasing with climate change” byHauvermale et al. submitted to Plants
The paper describes the production of novel antibodies against wheat amylases. These are enzymes are responsible for deteoriation fo wheat quality, if expressed at the wrong time, for example in late stages when wheat grain gets wet before the harvest. Current methods to asses the quality are error prone and a immunobased method would help to resolve these issues.
The paper is well written and the results sound. A few changes suggestions, comments are listed below, which might be beneficial to incorporate in the manuscript.
Introduction:
P2 line 60 The presence…..The information in that sentence is already spelled out before (line 39). It feels just a repetition here.
Results and Discussion:
Line 108 TaAmy. The abbreviation is explained in the methods as wheat sort, but not yet here. It would be good to spell it out either here or somewhere in the sentence below.
P3 Line 121 Though spelled out in the figure below, I would suggest to spell out DPA once in the main text as well.
Line 187 and Figure 4:
It would be good to maybe list/write out the actual sequences used for the production of the antibodies. They are in red in the figure, but that is not quite so visible. They could be written extra in the figure. It is not quite clear to me what the antigenic determinants in bold mean? Are they of any use? These are not sequences used for antibody production. For what are they important?
The generated antibodies, as well the peptides used to generate them are all named Tamy1 with a letter. I think, if taken into account that the isoforms of the amylase are Amy 1-4 and Tamy1-C is broad specific, the naming of the AB’S and the peptides as Tamy1-A/B/C can be confusing. Maybe the naming could be changed or just remove the 1 from the AB naming?
How have the dilutions be chosen? !/100 for two antibodies and 1/500 for the other.
From the Figure 6a it looks that Tamy1-C mostly recognize Tamy1 and not so much Tamy 2 and 4. Has an estimation of the Kd values been done? However, also alphaHIs shows differences. Is the load the same for the recombinant proteins on the gel?
Author Response
Reviewer 1 Comments and Suggestions for Authors (AUTHOR RESPONSES IN BOLD)
This is a review for the manuscript titled “Development of novel monoclonal antibodies to wheat Al-Pha-Amylases associated with gran quality problems that are increasing with climate change” byHauvermale et al. submitted to Plants
The paper describes the production of novel antibodies against wheat amylases. These are enzymes are responsible for deteoriation fo wheat quality, if expressed at the wrong time, for example in late stages when wheat grain gets wet before the harvest. Current methods to asses the quality are error prone and a immunobased method would help to resolve these issues.
The paper is well written and the results sound. A few changes suggestions, comments are listed below, which might be beneficial to incorporate in the manuscript.
Introduction:
P2 line 60 The presence…..The information in that sentence is already spelled out before (line 39). It feels just a repetition here.
Response 1: Thank you for catching this – the repetition has been removed.
Results and Discussion:
Line 108 TaAmy. The abbreviation is explained in the methods as wheat sort, but not yet here. It would be good to spell it out either here or somewhere in the sentence below.
Response 2: This has now been defined in the section header (line 165)
P3 Line 121 Though spelled out in the figure below, I would suggest to spell out DPA once in the main text as well.
Response 3: DPA has now been defined at first use (line 178).
Line 187 and Figure 4:
It would be good to maybe list/write out the actual sequences used for the production of the antibodies. They are in red in the figure, but that is not quite so visible. They could be written extra in the figure. It is not quite clear to me what the antigenic determinants in bold mean? Are they of any use? These are not sequences used for antibody production. For what are they important?
The generated antibodies, as well the peptides used to generate them are all named Tamy1 with a letter. I think, if taken into account that the isoforms of the amylase are Amy 1-4 and Tamy1-C is broad specific, the naming of the AB’S and the peptides as Tamy1-A/B/C can be confusing. Maybe the naming could be changed or just remove the 1 from the AB naming?
Response 4a: We chose to provide all information used to generate peptide for antibody production. All bold sequences (black and red) were predicted to be antigenic. The red peptides were those chosen for antibody production. This has been added to the figure legend (line 325) to add clarity.
Response 4b: The authors appreciate the reviewer’s suggestion but would prefer to keep the naming framework as all of the original antibody peptides were designed from TaAMY1 sequences.
How have the dilutions be chosen? !/100 for two antibodies and 1/500 for the other.
Response 5: Antibody dilutions were chosen based on empirical testing - serial dilutions of antibodies and targets.
From the Figure 6a it looks that Tamy1-C mostly recognize Tamy1 and not so much Tamy 2 and 4. Has an estimation of the Kd values been done? However, also alphaHIs shows differences. Is the load the same for the recombinant proteins on the gel?
Response 6: The samples loaded on the gel were from total protein extracts and estimated concentration for each was based on Braford assay. The loading of recombinant proteins was equal in terms of volume of E.coli lysate, but not protein. To compensate for this and to enable readers to assess the loading differences, we included the Coomassie blue stained gel panel at the bottom of figure 6a. An estimation of Kd values has not been performed.
Reviewer 2 Report
Comments and Suggestions for Authors
The work is important and comprehensive, but the idea that leads to the goal of the research is not always refined in the publication.
The paper presents the method used (falling number) and the proposed one in parallel, but there is no data comparison between these methods ("... The Hagberg-Perten Falling Number (FN) method is the industry standard for measuring α-amylase activity in wheatmeal. However, FN does not directly detect α-amylase and has major limitations. Developing α-amylase immunoassays would potentially enable early, accurate detection regardless of testing environment...")
Not always used sufficient citation and spectrum of literature sources; for example, in the literature are and other gradations of FN (e.g. Perten and other...), which also should be presented....
In the Materials and methods - insufficiently described growing conditions, also absence of data of humidity, temperature regime.... Also, insufficient explanation of varieties and their choices…
Question - why statistical methods are not presented at the end of the methodology; it is not clear how breeds (varieties) are reflected in the calculations? - it is necessary to explain this and under the graphs....
The scope of the conclusions is too large - it is more typical for the discussion – please to rewrite. The abstract must also be refined.
Some observations on language style were marked and presented in the comments of the manuscript…. Also found repeated text.... Note on formatting - Figures cannot be placed in the middle of a paragraphs.
That to improve the manuscript, there are uploaded file with the most of review points – they are marked and have comments.

Author Response
Reviewer 2 Comments and Suggestions for Authors (AUTHOR RESPONSES IN BOLD)
- The paper presents the method used (falling number) and the proposed one in parallel, but there is no data comparison between these methods ("... The Hagberg-Perten Falling Number (FN) method is the industry standard for measuring α-amylase activity in wheatmeal. However, FN does not directly detect α-amylase and has major limitations. Developing α-amylase immunoassays would potentially enable early, accurate detection regardless of testing environment...")
Response 1: We appreciate the reviewer’s comments. Direct comparisons between the FN methods and other rapids tests including previous developed immunoassays are described in the literature (citations 7, 29, and 30) and set a precedent for use. In the manuscript we describe that validation of new antibodies that can be used in previously described technology and as “a potential way” in the future for early detection. The validation of the antibodies described was the first step in developing of the immunoassays (lines 161-163).
- Not always used sufficient citation and spectrum of literature sources; for example, in the literature are and other gradations of FN (e.g. Perten and other...), which also should be presented....
Response 2: We acknowledge that additional should be added. We have gone back and added citations throughout the manuscript based on the reviewer’s comments. In total, there are 23 new citations including the one listed above.
- In the Materials and methods - insufficiently described growing conditions, also absence of data of humidity, temperature regime.... Also, insufficient explanation of varieties and their choices…
Response 3: The information about glass house temperatures, and information about sources of other seeds has been added in lines 769-828.
- Question - why statistical methods are not presented at the end of the methodology; it is not clear how breeds (varieties) are reflected in the calculations? - it is necessary to explain this and under the graphs...
Response 4: Statistical methods are given in lines 920-924. The fact significance was evaluated by ANOVA was added to the Figure 1, 2, and 3 legends. The only variety examined in Figures 1-3 was Chinese Spring. This is indicated in the legends.
- The scope of the conclusions is too large - it is more typical for the discussion – please to rewrite. The abstract must also be refined.
Response 5: We appreciate the reviewer’s comment and have revised the Conclusion section as requested. The first two paragraphs if the Conclusion were moved to the beginning of the Discussion section. Small refinements were made to the abstract, but with the 200 word limit it was not possible to add to it.
- Some observations on language style were marked and presented in the comments of the manuscript…. Also found repeated text.... Note on formatting - Figures cannot be placed in the middle of a paragraphs.
Response 6: We would like to note that many of the issues with formatting were not on the version we submitted, the document format was modified after submission presumably by the editorial staff. We apologize that the reviewers had to cope with this. We have correct the issues caused, corrected the title, moved all figures from the middle of paragraphs, and removed repeated text.
- That to improve the manuscript, there are uploaded file with the most of review points – they are marked and have comments.
Response 7: Thank you for your comments, we made revisions (marked with “track-changes”) as requested.